# Modeling Distributed MQTT Systems Using Multicommodity Flow Analysis

**Pietro Manzoni** [1,*] **, Vittorio Maniezzo** [2] **and Marco A. Boschetti** [3]

1   Department of Computer Engineering, Universitat Politècnica de València, 46022 Valencia, Spain
2   Department of Computer Science and Engineering, University of Bologna, 47521 Cesena, Italy;
    vittorio.maniezzo@unibo.it
3   Department of Mathematics, University of Bologna, 40126 Bologna, Italy; marco.boschetti@unibo.it
*   Correspondence: pmanzoni@disca.upv.es

**Abstract:** The development of technologies that exploit the Internet of Things (IoT) paradigm has led to the increasingly widespread use of networks formed by different devices scattered throughout the territory. The Publish/Subscribe paradigm is one of the most used communication paradigms for applications of this type. However, adopting these systems due to their centralized structure also leads to the emergence of various problems and limitations. For example, the broker is typically the single point of failure of the system: no communication is possible if the broker is unavailable. Moreover, they may not scale well considering the massive numbers of IoT devices forecasted in the future. Finally, a network architecture with a single central broker is partially at odds with the edge-oriented approach. This work focuses on the development of an adaptive topology control approach, able to find the most efficient network configuration maximizing the number of connections and reduce the waste of resources within it, starting from the definition of the devices and the connections between them present in the system. To reach the goal, we leverage an integer linear programming mathematical formulation, providing the basis to solve and optimize the problem of network configuration in contexts where the resources available to the devices are limited.

**Keywords:** publish/subscribe; Internet of Things; integer linear programming

## 1. Introduction

The Internet of Things (IoT) is a global network of connected devices, people, and processes, all of which collect and share data about how they are used and the environment around them. A timely analysis of the data coming from the IoT infrastructure is crucial in transforming it into knowledge that can add value to the application domain.

The information generated by IoT devices is typically sent to servers hosted in the cloud that can be far away. Li et al. [1] showed that the average round-trip time from various geographically distributed points to their optimal Amazon EC2 instances is 74 ms. To this transfer time, we should add the latency of the first wireless hop and the possible temporary connection failures, another major problem that interferes with developing critical applications or applications with real-time requirements. In [2], Bonomi et al. proposed the term "fog computing", which consists of a multilevel hierarchy of nodes spanning from the cloud to IoT devices. Edge or fog computing allows bringing AI-based IoT solutions in areas where connectivity is scarce and, in general, resources are limited, for example, in rural or remote areas. Consider, for example, the so-called TinyML solutions [3], a fast-growing field of machine learning technologies capable of performing on-device data analytics at extremely low power, typically in the mW range.

Therefore, a transition from a centralized, cloud-based architecture to an interoperable and decentralized dynamic IoT architecture looks promising. This transition will allow achieving highly efficient and responsive services by locating the data processing close to the data source, in the edge. However, current IoT infrastructures are not ready for

this move towards decentralization. The main challenge is how to include fog computing features in current infrastructures, ensuring the easy-to-use and high availability of current infrastructures, as well as what additional benefits this new paradigm can bring to IoT [4].

Publish/subscribe (Pub/Sub) is a widely used communication pattern in IoT applications. This pattern involves the publisher and the subscriber relying on a message broker that transfers messages to the subscribers. Although Pub/Sub is based on earlier design patterns like message queuing and event brokers, it is more flexible and scalable since Pub/Sub enables the movement of messages between different components of the system without the components being aware of each other's identity. Various messaging frameworks follow Pub/Sub's topic-based publishing and subscribing philosophy, like Apache Kafka or RabbitMQ. There are also efforts to provide this paradigm on top of Information-Centric Networking (ICN) architectures, like the Named-Data Networking (NDN).

This work focuses on the Message Queue Telemetry Transport (MQTT), an OASIS (Organization for the Advancement of Structured Information Standards) protocol for messaging between IoT devices that follow the Pub/Sub paradigm. MQTT aims to transport messages between devices requiring a small code footprint and limited network bandwidth. An MQTT client is any device, from a microcontroller to a server, that connects to an MQTT broker to exchange messages. The communication follows a topic-based publish/subscribe pattern with a broker acting as messages dispatcher. The broker is a central entity handling clients' connections, clients' subscriptions, and data publishing on specific topics. In this way, the broker allows decoupling data generation and consumption both in space and time, removing direct communication between the publisher of the data and the receiver (subscriber). This aspect, combined with the protocol simplicity at the client-side and the support for reliability and quality of service (QoS), makes MQTT an ideal candidate for resource-constrained applications.

The centralized structure of MQTT, but in general of Pub/Sub systems, has anyway drawbacks:

1. The broker is typically the single point of failure of the system: no communication is possible if the broker is unavailable.
2. It does not scale well considering the massive numbers of IoT devices forecasted in the future.
3. A network architecture with a single central broker is partially at odds with the edge-oriented approach we are considering where cloud services (including any broker instance) are moved to the edge, closer to the user devices.

New solutions are currently being tested to deal with these issues. In these solutions, all participating entities are connected via a sequence of publishers and subscribers linked to topics. Choosing the proper infrastructure for message broker implementation is crucial; otherwise, scaling can be hindered, and reliability issues may appear.

There are several design considerations for choosing a suitable message broker infrastructure; some of them are latency, bandwidth, message handling, service availability, service reliability, and security. A possible solution foresees the cooperation of multiple distributed MQTT brokers, acting as a single entity. Distributed brokers are deployed on different machines and connected over a network. The result is a single logical broker that ensures high scalability, replication, elasticity, and resiliency to failures. Specifically thought for being in cloud-based enterprise datacenters, many commercial brokers (i.e., EMQX, HiveMQ, and others) already provide message distribution and clustering capabilities. Clusters of brokers ensure publishing and subscriptions forwarding between the nodes along with other advanced features, such as broker discovery or failure recovery. Consequently, the lightweight principle of MQTT goes lost, often making communication overhead between the brokers non-negligible or increasing latency. This is unfavorable in an IoT scenario where deployments are often in constrained or frugal environments with brokers located at the network's edge.

Other brokers, like the widespread open-source Mosquitto implementation, use a bridging mechanism to exchange publications among distributed brokers. Bridging allows interconnecting brokers and distributing publications on specific topics among them, directly exploiting MQTT primitives. On the one hand, bridging makes the system simpler and entirely MQTT-based; however, it generally relies on a static configuration that may not scale in complex environments or cause message loops if not configured correctly. Such fragmented approaches may lead to noticeably different system performance based on the environment characteristics, i.e., underlying topology, network latency, clients distribution, etc.

This work takes a more theoretical approach to the overall problem and describes a distributed MQTT broker system. The problem can be cast as a network design problem, whose goal is to find the network configuration that can maximize the number of messages relayed to subscribers, or equivalently minimize the number of requests that are not satisfied by the network. In particular, we are focusing on the situation where links among brokers are bandwidth limited, for example, due to LPWAN technologies like LoRa.

The paper is structured as follows. Section 2 offers some of the relevant works in the area of this paper. Section 3 formally describes the studied problem presenting the first formulation. Section 4 illustrates an alternative formulation for the problem and Section 5 presents a computational validation. Finally, Sections 6 and 7 contain the future works and the provisional conclusions of this work.

## 2. Related Works

The research area of message dissemination in distributed generic pub/sub system has been very active in the last 20 years. Most works focus on the development of efficient and scalable routing algorithms to create topic-based dissemination trees (in the form of multicast groups) that cover only the subscribers matching a particular topic [5–9]. No specific broker implementation is considered in such works, and the overlay broker topology is assumed to be known. Only very recently, motivated by the protocol's popularity, some attention has been given to the problem of interconnecting MQTT-specific brokers [10]. Some works focused primarily on vertical clustering, where the single broker is replaced by many virtualized broker instances running behind a single endpoint, typically a load balancer [5,11]. These approaches introduce the concept of multiple brokers cooperating, although the broker cluster is seen as a single centralized entity from clients' perspectives. Pure MQTT broker distribution is introduced in Banno et al. in [12]: authors propose ILDM (Internetworking Layer for distributed MQTT brokers), where heterogeneous brokers are interconnected through specific nodes, placed between clients and brokers. Message distribution is obtained with publication flooding, but the underlying network of ILDM nodes is assumed to be already loop-free. Furthermore, no automatic mechanisms for broker failure recovery are present. In [13,14], authors also propose interconnecting MQTT brokers, with the possibility of dynamically changing the topology configuration at run time through specific MQTT messages transmitted by a centralized trusted entity.

On the same line, the work in [15] creates a broker network and uses an external monitoring agent to check the status of each broker. Clients are connected to brokers through local gateways: upon any change in the broker configuration (broker failure, increase in latency, etc.) the gateway reconnects the client to a new broker, according to the information retrieved by the monitoring agent. This approach enables client mobility, dynamic broker provisioning, and broker load balancing. In this work, we aim to achieve the same result by a completely decentralized architecture.

## 3. Problem Description

We consider a scenario where several sensors are distributed in an area producing data tagged according to their content according to a hierarchy of type *arg/subarg/subsubarg/...* (e.g., Bologna/station/humidity). The sensors are connected to clients responsible for collecting data from them and publishing them in the network. The network consists of

various clients and intermediate brokers accountable for connecting the clients using the MQTT protocol (see [16] for an example architecture).

Clients behave either as publishers or subscribers. The clients accessing the network can connect to one or many of the actual brokers and, through them, publish and receive data. The data, as well as the associated tags (topics), are characterized by the bandwidth required for their transmission in the network. In order to request a type of data, a client must subscribe to the specific topic of interest, a subscription that also admits wildcards (e.g., Lecco/humidity/# means any data on the humidity in Lecco).

All the existing connections in the network, both between client and broker and between broker and broker, have limited bandwidth, like for example in the case of a LPWAN link (e.g., [17]). The application layer protocol used by the network components to communicate with each other is the MQTT protocol. There is also the possibility that some nodes that make up the network are connected to the Internet, a case that results in the presence of essentially unlimited outbound bandwidth, but the connection with the Internet is not explicitly considered in the model presented in this paper.

Brokers have a limit on the allowed connections with other brokers or clients, and when the incoming requests reach the limit, a broker cannot accept further connections and rejects them.

Clients are not limited to connecting to only one broker but can choose to connect to any of the visible brokers, changing these links dynamically based on the load of connections a broker is subjected to. A network example is shown in Figure 1. In this case, client1 sees broker1 and broker2, client2 sees broker1 and broker4 and client3 see broker4 and broker7. If one of the brokers became saturated, a client might decide to open a connection to another of the brokers accessible by it.

The network can also change dynamically, with the possibility of a node disappearing or new ones appearing, just as connections between nodes can appear or disappear.

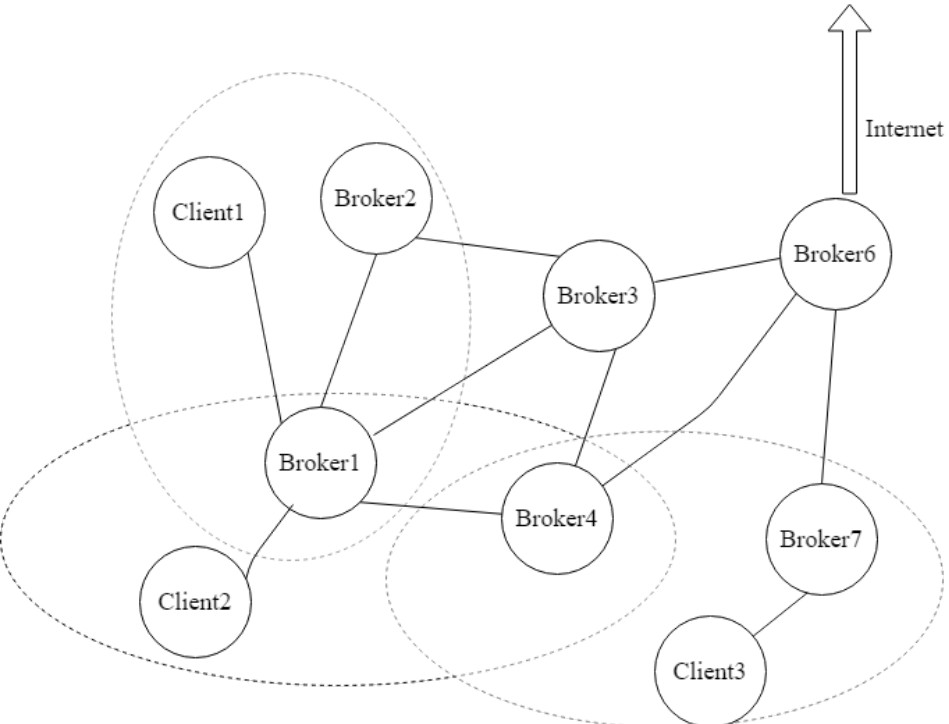

**Figure 1.** A simple example showing the relations among clients and brokers.

Brokers do not store data but only the tags requested by the clients or by other brokers connected to them. They are responsible for forwarding data compatible with the stored requests. In addition, brokers can themselves request data from other nodes; in particular, nodes connected to the Internet can request from other nodes the data that needs to be

forwarded over the network. In this way, dynamic paths between data producers and externally connected nodes will emerge.

Summing up, the problem can be described as a network design problem, whose goal is to find the network configuration that maximizes the number of connections forwarded to the subscribers, thus minimizing the number of requests that are not satisfied by the network. We will denote this problem the Publish/Subscribe network design Problem, P/SP for short.

### 3.1. A First Mathematical Formulation

To the best of our knowledge, no mathematical formulation has been so far proposed to model the problem of publish/subscribe network design. Here we propose and computationally validate one linear combinatorial optimization formulation and hint at a second formulation that could be of interest.

The problem to be optimized is an extension of a standard max-flow min-cost multicommodity flow problem [18], whose graph can be described using the following elements:

- $K$ is the set of commodities transmitted by the network, commodities that in our formulation correspond to the topics made available by the system.
- $S$ is the set of source nodes, in our problem it represents the set of clients that publish data related to each topic. Each commodity $k$ originates in a single client, thus in a subset $S^k \subseteq S$ with $|S^k| = 1$ (we keep the subset notation for possibly extending the model to multiple origins).
- $T$ is the set of destination nodes, in our problem it represents the set of clients that subscribe to the different topics. Each commodity $k$ can be requested by a set of clients $T^k \subseteq T$.
- $B$ is the set of intermediate nodes, in our problem it represents the set of brokers present in the network.
- $A$ is the set of arcs present in the graph. In our problem, it represents the connections client-broker and broker-broker. Note that arcs are assumed to be directed, thus edges are represented by pairs of arcs. Both arcs and edges are weighted by their associated bandwidth.

The publish/subscribe network design problem cannot be directly modeled as a multicommodity network design problem because the commodities generated by the sources could be required by multiple destinations, and in this case, data are to be generated only once but possibly duplicated and sent toward multiple destinations by the brokers used along the paths. Thus, the data flow exiting from the sources is not equal to the sum of the flows reaching the destinations, but possibly much smaller, contrary to multicommodity flow assumptions.

In our formulation, we further postulate that commodity flows cannot be split and recombined at destination and that sources and sinks are not brokers for their respective commodities, though sinks can be brokers for commodities they do not request. Sources have dedicated nodes with no entering arcs, while sinks have no exiting arcs for their commodity but require no dedicated node.

The problem is a maximization problem, we want to maximize the number of satisfied requests. The model formalizes this in a standard maximum flow minimum cost structure, where among all alternative feasible flow distributions that achieve the maximum request satisfaction objective, that of minimum cost is chosen. This permits to structurally make use of *costs* associated with arcs, a possibility that enables the Lagrangian optimization presented in Section 3.2. The pricing of arcs can easily turn the maximization problem into the problem of minimizing the number of unsatisfied requests: it is enough to add high-cost dummy arcs entering the sink nodes, that will be used only when no other feasible option can be found to satisfy their request.

Figures 2 and 3 present a very simple graph model of a publish/subscribe network. We have two publishing clients and one subscriber. The first publisher, S1, originates commodity (topic) c1, the second publisher, S2, originates commodity c2. There is only one

subscriber node, T, that subscribes to both topics. Note that edges among brokers can be used in both directions, while arcs from sources and to sinks are directed.

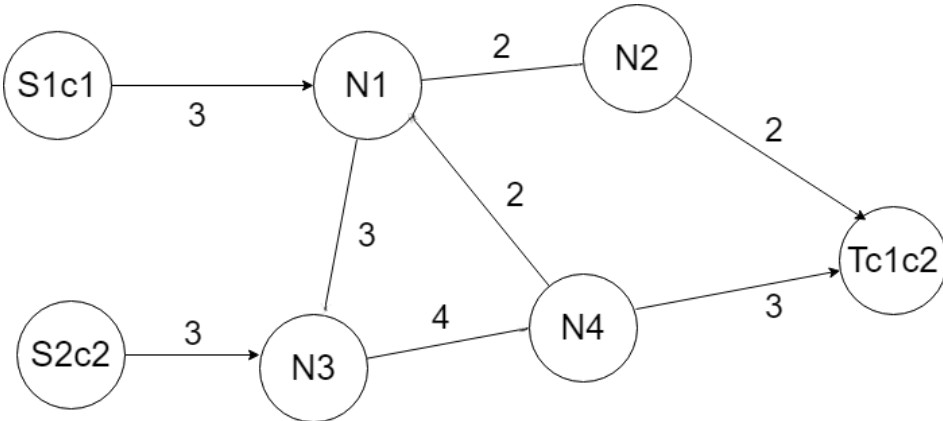

**Figure 2.** Flow network with multiple commodities.

Figure 3 shows the dummy arcs that are used to convert the problem from max flow min cost to minimum cost only. We remark that these arcs will be relaxed in a later phase of the procedure.

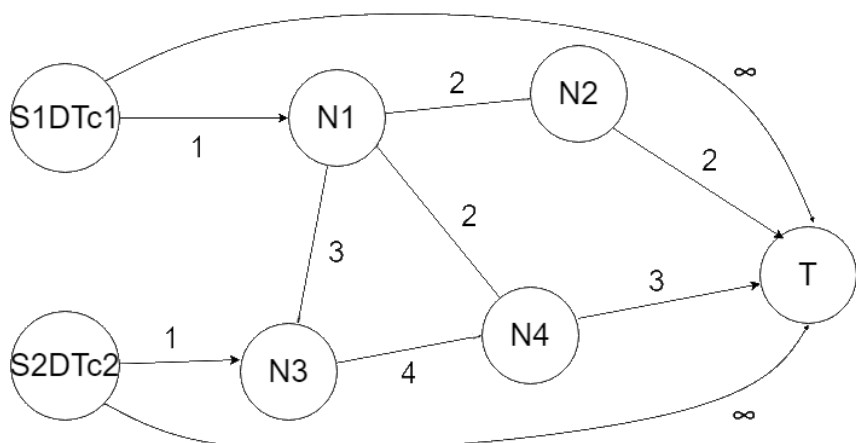

**Figure 3.** Min cost equivalent to max flow min cost network of Figure 2.

Based on the elements so far introduced, a first integer linear formulation for problem P/SP is the following one. We define:

- $x_{ij}^k$ as the *integer* variable that denotes how many clients will be served by the data flow transmitting commodity $k \in K$ using the arc $(i, j)$, i.e., how many data paths connect the publisher of $k$ and single subscribers going through arc $(i, j)$.
- $\xi_{ij}^k$ as the *binary* variable that takes value 1 if commodity $k \in K$ is transmitted along arc $(i, j)$.
- $cap_{ij}$ to be the overall capacity of edge $\{i, j\}$, accounting both for arc $(i, j)$ and $(j, i)$.
- finally, $c_{ij}$ to be the penalty that is paid if the arc $(i, j)$ is used.

The mathematical formulation, denoted as *F*1 is as follows:

$$\textbf{minimize}\, z_{F1} = \sum_{k \in K} \sum_{i \in N} \sum_{j \in \Gamma_i^{-1}} c_{ji} x_{ji}^k \tag{1}$$

**subject to**

$$\sum_{j \in \Gamma_t^{-1}} x_{jt}^k = 1 \qquad\qquad t \in T^k, k \in K \qquad\qquad (2)$$

$$\sum_{j \in \Gamma_i^{-1}} x_{ji}^k = \sum_{j \in \Gamma_i} x_{ij}^k \qquad\qquad i \in B, k \in K \qquad\qquad (3)$$

$$M\xi_{ij}^k \geq x_{ij}^k \qquad\qquad (i,j) \in A, k \in K \qquad\qquad (4)$$

$$\sum_{k \in K} a^k (\xi_{ij}^k + \xi_{ji}^k) \leq cap_{ij} \qquad\qquad i \in N, j \in \Gamma_i \qquad\qquad (5)$$

$$x_{ij}^k \geq 0 \text{ and integer} \qquad\qquad (i,j) \in A, k \in K \qquad\qquad (6)$$

$$\xi_{ij}^k \in \{0,1\} \qquad\qquad (i,j) \in A, k \in K \qquad\qquad (7)$$

where $N = S \cup B \cup T$ and $\Gamma_i$ and $\Gamma_i^{-1}$ denote the sets of endnodes, other than $i$, of arcs outgoing and incoming to node $i$, respectively.

Equation (1) represents the objective function of our problem. To optimize our problem, the costs $c_{ij}$ will have a non-negative value for arcs $(i,j)$, $i \in S$, $j \in B$, and 0 otherwise. In this way, the goal will be to minimize the cost of the internal flow in the network and penalize the flow that passes directly on dummy arcs between source and destination nodes.

Equation (2) represent the condition that every destination of commodity $k$ must receive the commodity.

Equation (3) represent the constraint that every commodity entering a broker node $i$ must also exit the node, and models the constraint of continuity and conservation of flow within the network. Equation (4) relates the variables $x_{ij}$ and $\xi_{ij}$, forcing each $\xi_{ij}$ to be 1 if the corresponding arc has a nonzero flow for commodity $k$.

Finally, Equation (5) represents the capacity constraint on the arcs. Finally, Equations (6) and (7) represent the integrality constraints on the decision variables.

Formulation $F1$ can be decomposed on the contribution of every single node. In fact, for each destination node $t \in T$ we have

$$\textbf{minimize } z_T = \sum_{k \in K} \sum_{j \in \Gamma_t^{-1}} c_{jt} x_{jt}^k \qquad\qquad (8)$$

**subject to**

$$\sum_{j \in \Gamma_t^{-1}} x_{jt}^k = 1 \qquad\qquad t \in T^k, k \in K \qquad\qquad (9)$$

$$M\xi_{it}^k \geq x_{it}^k \qquad\qquad i \in \Gamma_t^{-1}, k \in K \qquad\qquad (10)$$

$$\sum_{k \in K} a^k \xi_{it}^k \leq cap_{it} \qquad\qquad i \in \Gamma_t^{-1} \qquad\qquad (11)$$

$$x_{it}^k \geq 0 \text{ and integer} \qquad\qquad i \in \Gamma_t^{-1}, k \in K \qquad\qquad (12)$$

$$\xi_{it}^k \in \{0,1\} \qquad\qquad i \in \Gamma_t^{-1}, k \in K \qquad\qquad (13)$$

whereas, for each $i \in B$, we have:

$$\textbf{minimize } z_B = \sum_{k \in K} \sum_{j \in \Gamma_i^{-1}} c_{ji} x_{ji}^k \qquad\qquad (14)$$

**subject to**

$$\sum_{j \in \Gamma_i^{-1}} x_{ji}^k = \sum_{j \in \Gamma_i} x_{ij}^k \qquad\qquad k \in K \qquad (15)$$

$$M\xi_{ji}^k \geq x_{ji}^k \qquad\qquad j \in \Gamma_i^{-1}, k \in K \qquad (16)$$

$$\sum_{k \in K} a^k(\xi_{ij}^k + \xi_{ji}^k) \leq cap_{ij} \qquad\qquad j \in \Gamma_i^{-1} \qquad (17)$$

$$x_{ji}^k \geq 0 \text{ and integer} \qquad\qquad j \in \Gamma_i^{-1}, k \in K \qquad (18)$$

$$\xi_{ji}^k \in \{0,1\} \qquad\qquad j \in \Gamma_i^{-1}, k \in K \qquad (19)$$

The whole problem can thus be optimized in a fully distributed fashion, provided that routing decisions can be enforced downpath.

Unfortunately, problem P/SP and its distributed counterparts is *NP*-hard due to the arc capacity constraints, besides being based on the integer multicommodity flow problem, and we cannot, in general, expect to solve it within the characteristic time needed to operate a real-world P2P network.

We resort therefore to a distributed heuristic, specifically a matheuristic [19] that leverages the mathematical formulation we made available for our problem.

*3.2. Lagrangian Formulation*

Formulation *F1* is a particular multicommodity flow formulation with complicating capacity constraints. It comes natural to relax in a Lagrangian fashion the complicating constraints, obtaining a master problem defined on the binary variables and a subproblem implementing the specific multicommodity flow of interest.

In detail, we relax constraints Equation (5) and we insert them in the objective function with non negative penalties $\lambda_{ij}$, $(i,j) \in A$. The resulting formulation LP is as follows:

$$\textbf{minimize } z_{LP} = \sum_{k \in K} \sum_{i \in N} \sum_{j \in \Gamma_i^{-1}} (c_{ji} x_{ji}^k + \lambda_{ji} a^k(\xi_{ij}^k + \xi_{ij}^k))$$

$$- \sum_{(i,j) \in A} \lambda_{ij} cap_{ij} \qquad (20)$$

**subject to**

$$\sum_{j \in \Gamma_t^{-1}} x_{jt}^k = 1 \qquad\qquad t \in T^k, k \in K \qquad (2)$$

$$\sum_{j \in \Gamma_i^{-1}} x_{ji}^k = \sum_{j \in \Gamma_i} x_{ij}^k \qquad\qquad i \in B, k \in K \qquad (3)$$

$$M\xi_{ij}^k \geq x_{ij}^k \qquad\qquad (i,j) \in A, k \in K \qquad (4)$$

$$x_{ij}^k \geq 0 \text{ and integer} \qquad\qquad (i,j) \in A, k \in K \qquad (6)$$

$$\xi_{ij}^k \in \{0,1\} \qquad\qquad (i,j) \in A, k \in K \qquad (7)$$

$$\lambda_{ij} \geq 0 \qquad\qquad (i,j) \in A \qquad (21)$$

The constraint matrix now has a block structure and decomposes over the different commodities, the only linkage being the objective function. It is thus possible to solve separately for each commodity, identifying a Dijkstra tree rooted in the publisher and having leaves in the subscriber nodes.

Formulation LP paves the way to the design of a Lagrangian matheuristic [20,21], where the subproblem is solved to optimality and a suitable search method, in our case subgradient optimization, is used to try to get a feasible solution.

Formulation $F1$ actually contains the standard integer programming formulation of the Single Source Shortest Path problem (SSSP) [22] and formulation LP keeps it as a subproblem, therefore the solution of the subproblems is readily made by means of any code for the SSSP. Unfortunately, the Dijkstra algorithm is surely a suitable option for the general case, but it lends itself very poorly to a distributed implementation. However, distributed alternatives to Dijkstra for standard SSSP instances have been studied [23].

## 4. An Alternative Formulation of the Problem

As an alternative to the formulation presented in the previous section, which is based on SSSP, a second formulation can be proposed that tries to combine transmissions on fewer arcs, thus saturating even more their bandwidths.

The problem to be optimized is again a problem of type Multicommodity flow, whose graph is described as follows:

- $V$ is the set of nodes of the problem, which are partitioned into client nodes $V_c$ and broker nodes $V_b$, such that $V = V_c \cup V_b$;
- $A$ is the set of arcs present in the graph, which in our problem represent the connections client-broker and broker-broker;
- $K$ is the set of commodities present in the network, commodities which in our problem model the topic present in the system.

Again, it is necessary to add arcs connecting the sources with the destinations, otherwise, there are not necessarily minimum cost paths connecting every subscriber, and the use of these dummy arcs by the solution will indicate the impossibility of optimizing the problem with a given network configuration.

The notation used in this formulation is:

- $\xi_{ij}^k$ is a binary variable that takes value 1 if the commodity $k \in K$ is transmitted by arc $(i, j)$;
- $x_{ij}^k$ as the positive variable that quantifies how much of the bandwidth relative to commodity $k \in K$ will eventually be transmitted to the subscribers going through arc $(i, j)$.
- $cap_{ij}$ indicates the capacity of each arc $(i, j)$;
- $n_{max}$ is a parameter indicating how many times a commodity can be duplicated within each single broker node;
- finally, $c_{ij}$ is the penalty that is paid if the arc $(i, j)$ is used.

A possible mathematical formulation, denoted $F2$, is as follows:

**minimize**

$$z_{F2} = \sum_{k \in K} \sum_{i \in N} \sum_{j \in \Gamma_i^{-1}} c_{ji} \xi_{ji}^k \tag{22}$$

**subject to**

$$\sum_{j \in \Gamma_s} \xi_{sj}^k = 1 \qquad\qquad s \in S^k, k \in K \tag{23}$$

$$\sum_{j \in \Gamma_t^{-1}} \xi_{jt}^k = 1 \qquad\qquad t \in T^k, k \in K \tag{24}$$

$$n_{max} \sum_{j \in \Gamma_i^{-1}} \xi_{ji}^k \geq \sum_{j \in \Gamma_i} \xi_{ij}^k \qquad\qquad i \in B, k \in K \tag{25}$$

$$\sum_{j \in \Gamma_t^{-1}} x_{jt}^k = a^k \qquad\qquad t \in T^k, k \in K \tag{26}$$

$$\sum_{j \in \Gamma_i^{-1}} x_{ji}^k = \sum_{j \in \Gamma_i} x_{ij}^k \qquad\qquad i \in B, k \in K \tag{27}$$

$$a^k \xi_{ij}^k \leq x_{ij}^k \leq a^k |T^k| \xi_{ij}^k \qquad (i,j) \in A, k \in K \qquad (28)$$

$$\sum_{k \in K} a^k \xi_{ij}^k \leq cap_{ij} \qquad (i,j) \in A \qquad (29)$$

$$x_{ij}^k \geq 0 \qquad (i,j) \in A, k \in K \qquad (30)$$

$$\xi_{ij}^k \in \{0,1\} \qquad (i,j) \in A, k \in K \qquad (31)$$

Equation (22) represents the objective function of our problem. To optimize our problem, all coefficients $c_{ij}$ will initially take on positive values for each arc $(i,j) \in A$ and 0 otherwise. In this way, the goal will be to minimize the cost of the internal flow in the network by penalizing the dummy flow that passes directly between source and destination nodes.

Equation (23) represent the condition that every source of commodity $k$ must generate the commodity, while Equation (24) ensure that every destination of commodity $k$ must receive it.

Equation (25) represent the constraint that every commodity entering a broker node $i$ must also exit, and models the constraint of continuity and conservation of flow within the network.

Equation (26) ensure that all requests are satisfied (either by the network or through the dummy arcs), Equation (27) the continuity of the flows and Equation (28) link the two sets of variables, besides collectively ensuring that flows cannot be split.

Equation (29) represent the capacity constraint on the arcs. Finally, Equation (31) represent the integer constraints on the decision variables and, in particular, they tell us that they can only take on values 0 or 1.

## 5. Computational Validation

Formulation *F*1 has been prototypically implemented in order to validate the approach and determine the effectiveness of the Lagrangian heuristic when compared against the optimal solution of the tested instances.

All code was written in C# and made use, through a specific wrapper we devised [24], of a MIP solver of choice, currently either CoinMP or IBM Cplex; in the following results were obtained by using IBM Cplex ver. 12.5. The computational platform was a Windows 11 Intel I7 machine, running at 1.6 GHz and using 8 GB of RAM. We made no explicit effort to parallelize code on different cores.

A note on the code we used for subgradient optimization of the Lagrangian dual. In this work, we used a standard subgradient optimization code [25], and based the Lagrangian penalty update on the classical Polyak rule [26]. We are aware that this rule is based on global parameters, but this limitation can be bypassed without losing the guarantees on the solution convergence in a fully distributed implementation as shown in [20].

Since we have no references to previous attempts to solve this problem, we generated an instance testset, partially adopting some of the integer multicommodity flow instances proposed in [27], specifically of the "Reserve" dataset, which are instances arising in telecommunications and asking for optimally sizing reserve capacities on the arcs of an existing telecommunication network to face one-arc "catastrophic" failures. The test instances are available from [28], in JSON format.

Figure 4 shows the full graph of instance *tinyInstance*, available in JSON encoding from [28]. Black arcs refer to commodity 1, generated from node 4, while green arcs are for commodity 2, generated by node 5. Both source nodes have only outgoing arcs. Both commodities are requested by node 3, which has only incoming arcs. Nodes 0, 1, and 2 are broker nodes and are connected by arcs allowing both communication directions. Arcs (4,3) and (5,3) are the dummy, expensive arcs, included to ensure feasibility.

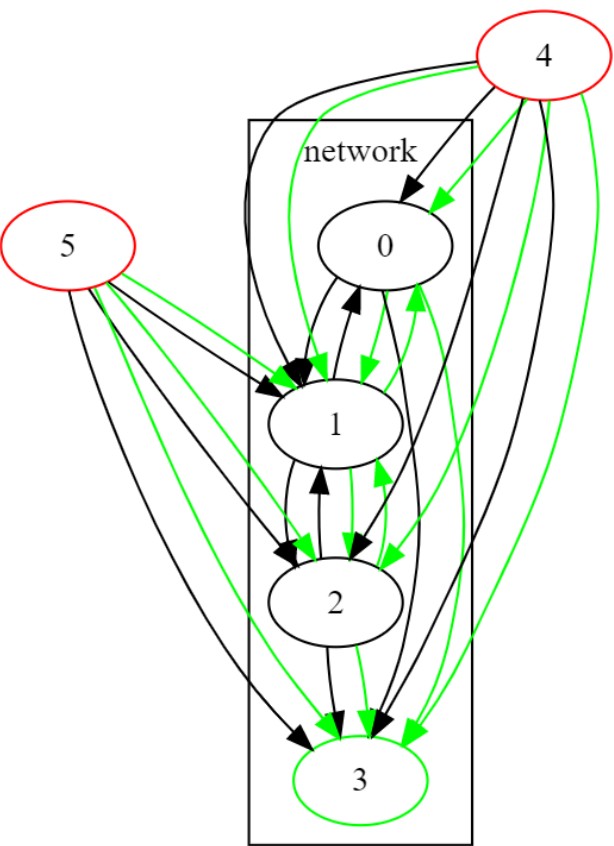

**Figure 4.** The full graph of instance TinyInstance.

The preliminary tests carried out so far were made over five instances of increasing complexity, whose characteristics are listed in Table 1. The table reports for each instance its conventional name, its total number of nodes, its total number of arcs, where the bidirectional ones have already been split into two opposite direction arcs, its number of topics (commodities), and its number of subscribers. In all cases, the number of publishers is equal to the number of topics and the number of brokers can thus be easily derived.

**Table 1.** Initial testset.

| Name | n.nodes | n.arcs | n.topics | n.subscr |
|---|---|---|---|---|
| tinyInstace | 6 | 13 | 2 | 1 |
| case2 | 11 | 23 | 3 | 2 |
| case3 | 20 | 57 | 6 | 3 |
| res0 | 14 | 28 | 5 | 6 |
| res5 | 75 | 165 | 31 | 24 |
| res7 | 71 | 244 | 31 | 23 |

We applied to each of these instances both the distributed MIP formulation T, B (see Equations (8)–(19)) and the relaxed Lagrangian formulation LP (see Equations (20) and (21)), with the subproblem solved by heap-based SSSP code.

To complete the computational validation of our approach we also adapted to our case a Simulated Annealing (SA) solution. As P/SP is a new problem, there are no proposals in the literature specific to it. We, therefore, considered the closest contribution we could find, a SA for the multi-source single-path multi-commodity network flow problem that explicitly mentioned telecommunication system applications [29]. Two main differences exist between the two problems; P/SP is more constrained in that it leaves no freedom in

choosing the bandwidth to allocate to the paths of each commodity (the $a_k$ variables of the SA in the literature) and rather than multiple sources we have multiple destinations for each commodity. We chose to modify minimally the original contribution, just so as to make it applicable to our problem. Bandwidth allocation becomes a fixed cost for unmet capacity constraints, while the multiple destination objective does not impact the solution strategy. Deeper adaptations are clearly possible to enhance the SA effectiveness, but this would lead the result away from the literature proposal we want to compare against. We leave this research effort to a possible interested reader.

Table 2 gives a summary of our current results. The MIP solver was allowed to run for at most 600 s or until memory dictated to stop. Column t.MIP reports the time to optimality, which was achieved for all the reported instances.

The Lagrangian metaheuristic was able to find the optimal solution for all instances, too. We report the gap from optimality in column gap, while in column opt we put an asterisk if the heuristic itself was able to prove the optimality of the obtained solution, a possibility inherent to Lagrangian metaheuristics. This proof was achieved only for the small instances, while for the bigger ones the code could find the optimal solution but not to prove its optimality.

The SA is reported only for reference, more adaptations are needed to make it effective on this problem. Given the stochastic nature of the algorithm, all results are relative to 5 runs on each instance. Column gap reports its average percentual gap from optimality, column n.opt the number of times, over 5, when the optimal solution was produced, and column t.SA the average CPU time needed to produce the best solution found at each try. When the code was consistently unable to produce feasible solutions, we report a n.a.

A few further remarks are in order here. The current code is definitely not optimized for speed, but it is to be considered a proof of concept of the effectiveness of the approach. The reported instances are all relatively easy, and in fact, contain only the smaller one of the "Reserve" dataset. It is not of interest for this stage of our research to attack more complex instances, rather to design a fully distributed implementation of the algorithm, where each node (each broker) makes local decisions and the globally optimal solution emerges. The Lagrangian approach was selected with this objective in mind, and the results reported here confirm that its use does not limit the effectiveness of the method eventually obtained.

**Table 2.** Results using the Lagrangian metaheuristic; an asterisk indicates that the heuristic itself was able to prove the optimality of the obtained solution.

| | MIP | | Lagr | | | SA | |
| --- | --- | --- | --- | --- | --- | --- | --- |
| Name | t.MIP | gap | opt | t.Lagr | gap | n.opt | t.SA |
| tinyInstance | 0.3 | 0 | * | 0.01 | 0 | 5 | 1.8 |
| case2 | 0.3 | 0 | * | 0.07 | 12 | 3 | 12.3 |
| case3 | 0.4 | 0 | * | 0.08 | 74 | 1 | 68.5 |
| res0 | 0.4 | 0 | * | 0.07 | 68 | 2 | 56.0 |
| res4 | 8.2 | 0 | | 0.39 | 174 | 0 | 73.4 |
| res5 | 67.9 | 0 | | 0.35 | 209 | 0 | 78.9 |
| res7 | 75.2 | 0 | | 1.21 | n.a. | 0 | n.a. |

Figure 5 shows the solution, identically obtained by both codes, on the instance of Figure 4. Broker 1 proves to be redundant for the simple requests of this instance.

Results of more interest are obtained on instance case2, which is more constrained on the available bandwidths. A notable feature, that sets the P/SP problem apart from common network flow problems, is the possibility given to brokers to multiplicate incoming data flows.

Instance case3 is yet more demanding and requires a more complex search space exploration. The result, obtained by the Lagrangian heuristic code, is presented in Figure 6.

It is noteworthy that several flow duplications are suggested (brokers 2, 4, 8, 10), but one broker is still deemed unneeded for the transmission requirements of this instance.

All instances so far presented permit to satisfy all transmission requests, as can be seen from the lack of arcs directly connecting publishers to subscribers. However, tight bandwidth limits force the use of different paths in order to connect the data sources to the respective consumers.

Instance case3 is the first one whose complexity causes the MIP and the Lagrangian approach to produce different results, albeit in this case still equivalent ones. Figure 7 shows the result obtained by running a MIP solver on the T-B formulation (it is unfortunate that we have little control over the layout of the graph produced by [30]). Both formulations leave a broker out and suggest several duplications, to be implemented in different brokers. The CPU time required to reach these solutions is still less than a second in both cases.

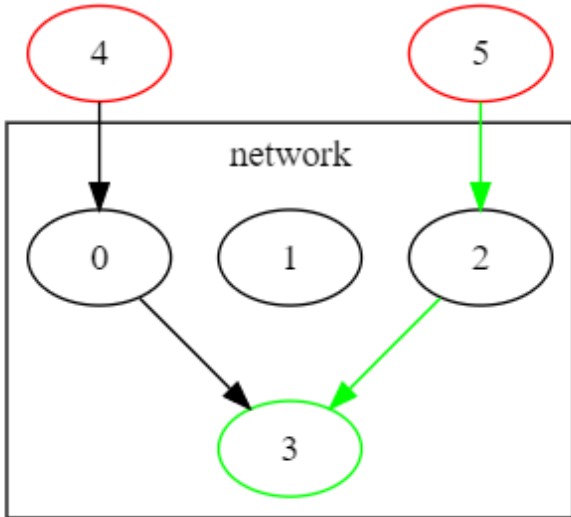

**Figure 5.** TinyInstance results.

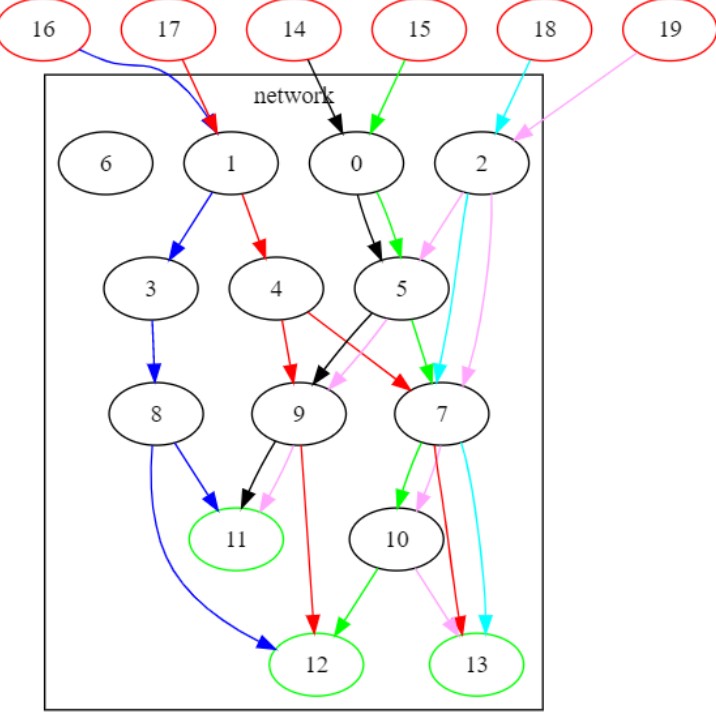

**Figure 6.** Instance case3 Lagrangian result.

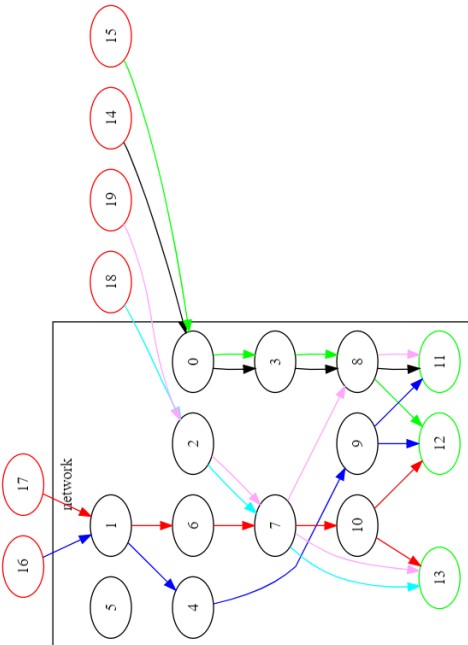

**Figure 7.** Instance case3 MIP result.

Moving to instances derived from real world transmission requests, we first faced the smaller one, res0. The result, obtained by both codes, is presented in Figure 8. The instance proves to be not particularly challenging, except for requiring to route flows on paths counting more than the minimum feasible number of arcs (see commodity 13) because of bandwidth constraints.

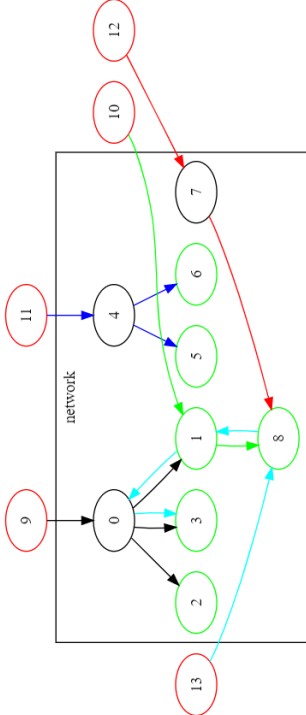

**Figure 8.** Instance res0 result.

Instances res5 and res7 are much bigger and permit us to appreciate the different computational requirements between exact and heuristic codes. We report in Figure 9 the solution of the Lagrangian heuristic for instance res7 as a suggestion to the complexity of

the faced task. In both cases, the solutions found were equivalent from the viewpoint of the objective function cost, but were different in structure. Clearly, there is no guarantee this to be the case also with other, possibly more complex, problem instances.

What was really different was the CPU time required to get to these solutions. In fact, while the Lagrangian heuristic took less than 10 s to produce a solution, the MIP solver needed more than an hour in both cases. However, we must note that this CPU time was mostly spent in trying to prove the optimality of the best found solution: the final solution was in fact found early in the search process.

A final remark about formulation *F*2. We implemented it in a non-distributed, preliminary version. In this setting, results are equivalent to those obtained by formulation *F*1, both in quality and in CPU time. We expect that a full computational test, on more instances and possibly on real-world publish/subscribe use cases, will differentiate the proposed alternatives.

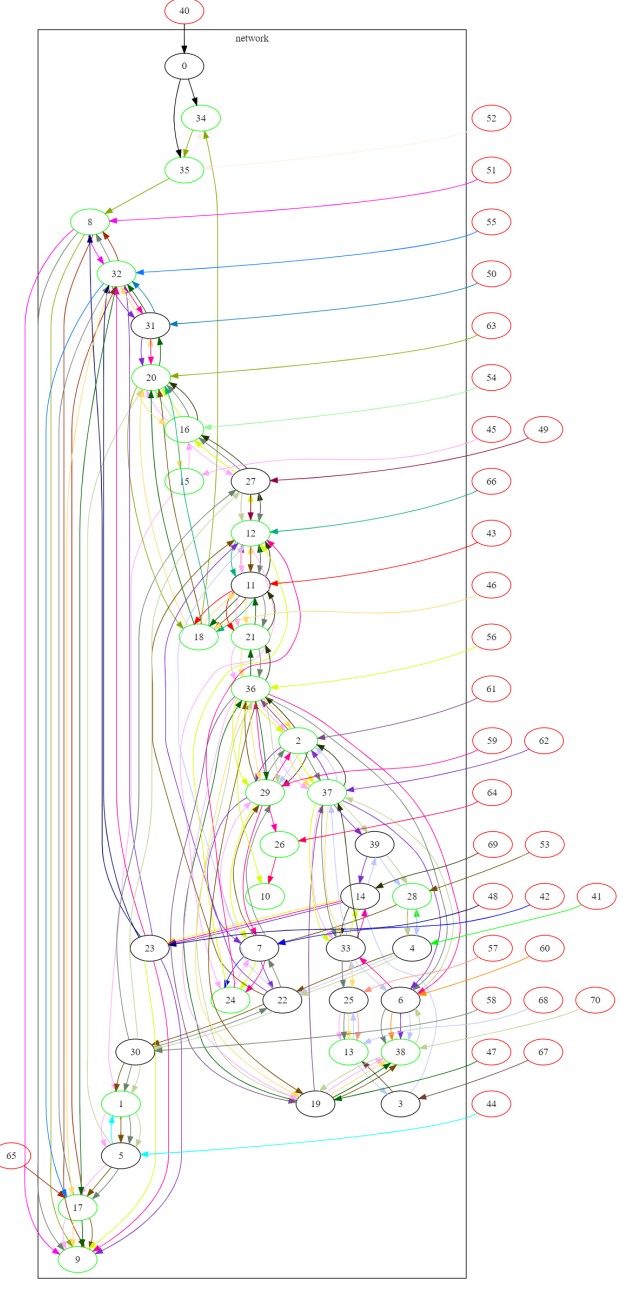

**Figure 9.** Result on use case res7 using the Lagrangian formulation.

## 6. Future Work

This work is still in its preliminary phase. We wanted to address the overall problem, even if partially, to understand the set of issues better to be dealt with. We formulate the problem for selecting the best routing configuration, and we provide insights on how this can be done in a distributed fashion. We are currently working on how this can be practically implemented, i.e., how clients and brokers exchange information to enable the proposed solution. This is a crucial part and requires more research. Finally, the various proposed formulations exhibit different computation times; we want to investigate how those times change as a function of the network scale and other systems-related parameters.

## 7. Conclusions

In this paper, we described the groundwork upon which to complete the design of communication protocol elements that will enable an adaptive, self-organizing design of a publish/subscribe service network. This result is to be achieved in steps, first designing a suitable optimization approach for communication flows routing, then distributing it over the network nodes so that optimization can be achieved asynchronously and in parallel, and finally identifying data that need to be exchanged among nodes in order to enable distributed optimization.

We reported here the preliminary results of the first and partially about the second of these steps. We extended classical multicommodity flow models in different ways, in order to account for the peculiarities of the P/SP problem and validated the new model both on artificial and on simple real-world derived instances. We also proposed and tested a possible distribution of the optimization models.

The proposed models are based on mixed-integer linear programming (MILP) formulations. Since the P/SP problem is NP-hard, these formulations are not expected to be able to efficiently cope with large-scale real-world instances, therefore we also proposed a Langrangian heuristic based on one of these formulations. Preliminary computational results are encouraging.

In a context where the IoT is becoming more and more present and used, having a way to autonomously optimize the communication networks used by this type of system plays a crucial role in their design. In addition, the study and application of MILP formulations allowed us to acquire a practical view on the use of computer science and programming with respect to real problems, highlighting how different computer science topics and techniques can be beneficial to one another and permit advancements of mutual interest.

Currently we are working on completing the computational test of the available codes and on the implementation of the distributed optimization modules on network nodes in an industrial P2P network simulation system. We expect to be able to provide factual data on the effectiveness of our distributed network design and on its level of resilience when facing network disruptions or varying data demands.

Finally, we like to mention a possible important extension of our model. As variables such as the transmission rate of different users have a high impact on the solution structure, and we do not want the solution to be fluctuating, a relevant research line considers robust optimization of the proposed solution.

**Author Contributions:** Conceptualization, P.M.; Data curation, M.A.B.; Formal analysis, V.M. and M.A.B.; Validation, V.M.; Writing—original draft, P.M.; Writing—review & editing, P.M., V.M. and M.A.B. All authors have read and agreed to the published version of the manuscript.

**Funding:** This work is derived from R&D project RTI2018-096384-B-I00, funded by MCIN/AEI/ 10.13039/501100011033 and "ERDF A way of making Europe", and by the European Union's Horizon 2020 research and innovation programme under grant agreement No. 101017861.

**Conflicts of Interest:** The authors declare no conflict of interest.

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
