# Peer review of "Modeling Distributed MQTT Systems Using Multicommodity Flow Analysis"

_electronics, doi:10.3390/electronics11091498_

Round 1

Reviewer 1 Report

This paper considers the optimization formulation of MQTT systems by applying the multicommodity flow framework. The solution can improve the performance of MQTT systems. However, even for a static network, the problem still takes a significant time to solve. A relaxed model is proposed and it can be solved in a reasonable time if the mobility of the nodes is not high. As a preliminary work, this paper shows the importance and the hardness of the problem.

Here are my questions:

  1. In line 276, it is mentioned that subgradient optimization is used to try to get a feasible solution of LP. Does the approach (primal? dual? primal-dual?) global asymptotic stable? Usually when the individual variables are important in the system (e.g., the transmission rate of different users in the utility maximization problem of TCP), we don’t want the solution to be fluctuating although the objective is optimized. Yet, it may not be easy to find a Lyapunov function so this may be a future direction in case the authors do not know the answer at this moment.
  2. Besides using MIP solvers, is there any reason not to compare with some simple heuristics such as simulated annealing or Markov approximation (log-sum-exp approximation for combinatorial optimization)? Simulated annealing may be used as a baseline for comparing the performance of the proposed heuristic LP.
  3. What is the specification of the devices used in the experiments (CPU, memory, etc.)? Does the solver use multiple cores?
  4. In line 155, why is it assumed that if a node can access the Internet, then that link has essentially unlimited outbound bandwidth? How does this assumption affect the formulation in this paper?

The followings are some minor comments:

  1. Line 84. “… publishes and subscriptions …”. The former one is a verb while the latter one is a noun. Do you mean “publications”?
  2. In the formulation of F1, it seems that $\Gamma_i$, $\Gamma_i^{-1}$, and $M$ are not defined.
  3. In the formulation of F2, Eqs. (30) and (31) did not mention which sets i, j, k belong to? (For consistency only, as all the other constraints include this information)
  4. Line 330. I believe it is C# instead of c# (capital letter).
  5. Lines 334 and 336. The double quotes seem not being typeset correctly. You may refer to Line 27 to see the difference.
  6. There are many undefined references. For example, in Lines 334, 337, 339, 371.
  7. Line 434 overflows.
  8. Reference 16. I think it should be pp. instead of p. for the page numbers.

Author Response

We thank the reviewers for their constructive remarks, that we included in the revised version of our manuscript as follows. 

>. . . 

>In line 276, it is mentioned that subgradient optimization is used to try to get a feasible solution of LP.  

>Does the approach (primal? dual? primal-dual?) global asymptotic stable?  

We included appropriate references to the theory supporting the method we implemented, mentioning also the convergence guarantees, at the beginning of the computational validation section. 

>Usually when the individual variables are important in the system (e.g., the transmission rate of  

>different users in the utility maximization problem of TCP), we don’t want the solution to be fluctuating  

>although the objective is optimized. Yet, it may not be easy to find a Lyapunov function so this may be  

>a future direction in case the authors do not know the answer at this moment. 

We mentioned this interesting further study possibility in the conclusions, that in our case translates in a study of a robust optimization method for our problem. This falls outside the scope of our contribution, but it surely represents a worthwhile further investigation topic. 

>Besides using MIP solvers, is there any reason not to compare with some simple heuristics such as  

>simulated annealing or Markov approximation (log-sum-exp approximation for combinatorial  

>optimization)? Simulated annealing may be used as a baseline for comparing the performance of the  

>proposed heuristic LP. 

Following the reviewer's suggestion, we implemented a SA, adapting to our problem the closest contribution we could find in the literature. As it turns out, the current implementation is no match for our method, but we acknowledge that it could be easy to improve the SA performance. This is, however, again not within the scope of our research, and we mentioned it as a further possible extension. 

We do not believe that Markov approximation can be made in any way competitive against the approach we advocate in the paper. 

>What is the specification of the devices used in the experiments (CPU, memory, etc.)? Does the solver  

>use multiple cores? 

We added these data. 

>In line 155, why is it assumed that if a node can access the Internet, then that link has essentially  

>unlimited outbound bandwidth? How does this assumption affect the formulation in this paper? 

We do not explicitly consider transmissions to/from the Internet in the model, we made it clear that this was just a comment for a possibility that could arise, probably with cable links that justify the assumption.  

>Line 84. “… publishes and subscriptions …”. The former one is a verb while the latter one is a noun.  

>Do you mean “publications”? 

Corrected 

>In the formulation of F1, it seems that $\Gamma_i$, $\Gamma_i^{-1}$, and $M$ are not defined. 

You are right, we took them for granted. Now we define them.

>In the formulation of F2, Eqs. (30) and (31) did not mention which sets i, j, k belong to? (For consistency 

> only, as all the other constraints include this information) 

Corrected

>Line 330. I believe it is C# instead of c# (capital letter). 

Corrected 

>Lines 334 and 336. The double quotes seem not being typeset correctly. You may refer to Line 27 to 

 >see the difference. 

Corrected 

>There are many undefined references. For example, in Lines 334, 337, 339, 371. 

Corrected 

>Line 434 overflows. 

Corrected

>Reference 16. I think it should be pp. instead of p. for the page numbers. 

That was automatically generated by bibtex – latex (with \bibliographystyle{plain}). I think this will be fixed in a later stage by the publisher

Reviewer 2 Report

The topic is interesting referring to the Message Queue Telemetry Transport (MQTT), an OASIS (Organization for the Advancement of Structured Information Standards) protocol for messaging between IoT devices that follow the Pub/Sub paradigm. MQTT aims to transport messages between devices requiring a small code footprint and limited network bandwidth.  In this context, the authors developed an adaptive topology control approach, able to find the most efficient network configuration maximizing the number of connections and reducing the waste of resources within it, starting from the definition of the devices and the connections between them present in the system.
The structure of the paper is correct, beginning with a good introduction, where the goals of the investigation are presented. In the next sections, the proposed approaches are explained based on mathematical modeling. Finally, the testing led the encouraging results.
Some observations can be made to improve the structure of the paper:
1. In the first part, the authors present various approaches from the literature. I think that a synthesis of the solutions proposed in the literature depending on the type of analysis, which highlights more clearly the advantages and disadvantages, is useful for readers. This synthesis can be given as a table.
2. Please check all text. There are some symbols [?] that appear and I think that a reference is missing between the brackets.
3. Details on the optimization process should be indicated in the tables.

Author Response

We thank the reviewers for their constructive remarks, that we included in the revised version of our manuscript as follows. 

>1. In the first part, the authors present various approaches from the literature. I think that a synthesis 

> of the solutions proposed in the literature depending on the type of analysis, which highlights more 

> clearly the advantages and disadvantages, is useful for readers. This synthesis can be given as a table. 

We considered about this possibility but we eventually decided  that a table might miss to properly present the interrelation among the described proposal and would not properly illustrate the sequence of the dependencies

>2. Please check all text. There are some symbols [?] that appear and I think that a reference is missing 

> between the brackets. 

Corrected

  1. Details on the optimization process should be indicated in the tables.

We added more detailed data, both on the algorithms and on the platform we ran them on.